# The Clinical Outcomes and Safety of Sacituzumab Govitecan in Heavily Pretreated Metastatic Triple-Negative and HR+/HER2− Breast Cancer: A Multicenter Observational Study from Turkey

**DOI:** 10.3390/cancers17091592

**Published:** 2025-05-07

**Authors:** Harun Muğlu, Kaan Helvacı, Bahadır Köylü, Mehmet Haluk Yücel, Özde Melisa Celayir, Umut Demirci, Başak Oyan Uluç, Gül Başaran, Taner Korkmaz, Fatih Selçukbiricik, Ömer Fatih Ölmez, Ahmet Bilici

**Affiliations:** 1Medical Oncology Department, Faculty of Medicine, İstanbul Medipol University, Istanbul 34810, Türkiye; mehmet.yucel@medipol.edu.tr (M.H.Y.); omerfatih.olmez@medipol.com.tr (Ö.F.Ö.); abilici@medipol.edu.tr (A.B.); 2Medical Oncology Department, Faculty of Medicine, Yüksek İhtisas University, Ankara 06520, Türkiye; kaan.helvaci@memorial.com.tr; 3Medical Oncology Department, Faculty of Medicine, Koç University, Istanbul 34010, Türkiye; bkoylu@ku.edu.tr (B.K.); fselcukbiricik@kuh.ku.edu.tr (F.S.); 4Medical Oncology Department, Faculty of Medicine, Acıbadem University, Istanbul 34752, Türkiye; melisacelayir84@gmail.com (Ö.M.C.); basak.uluc@acibadem.com (B.O.U.); gul.basaran@acibadem.edu.tr (G.B.); taner.korkmaz@acibadem.com (T.K.); 5Medical Oncology Department, Memorial Ankara Hospital, Ankara 06520, Türkiye; umut.demirci@memorial.com.tr

**Keywords:** sacituzumab govitecan, real-world data, metastatic breast cancer, triple-negative breast cancer, hormone receptor-positive breast cancer, Trop-2, antibody–drug conjugate

## Abstract

Sacituzumab govitecan (SG) is a novel antibody–drug conjugate used to treat advanced breast cancer, particularly in patients with a metastatic triple-negative (mTNBC) or metastatic hormone receptor-positive/HER2-negative (mHRPBC) subtype who have limited treatment options. Although clinical trials have shown its effectiveness, data from real-world clinical settings remain limited, especially for mHRPBC patients. In this multicenter retrospective study, we evaluated the outcomes of SG treatment in 68 patients from routine clinical practice in Turkey. Our findings show that treatment with SG resulted in similar progression and survival outcomes across two major breast cancer subtypes, with a manageable safety profile. This study aimed to provide real-world observational data describing the clinical outcomes and tolerability of SG in patients with mTNBC and mHRPBC. This is one of the first real-world studies to include mHRPBC and provides important insight into how this drug performs outside of clinical trials.

## 1. Introduction

Breast cancer remains the most frequently diagnosed malignancy in women worldwide. Despite significant progress in screening methods and systemic treatment options, breast cancer continues to result in substantial clinical challenges, particularly in its advanced stages [1,2].

Metastatic hormone receptor-positive, HER2-negative breast cancer (mHRPBC) constitutes approximately 70% of all cases and is typically managed with endocrine therapies in combination with targeted agents, such as cyclin-dependent kinase 4 and 6 inhibitors (CDK 4/6). However, resistance to hormonal treatments inevitably emerges in many patients, necessitating the use of chemotherapy in later lines. In the metastatic setting, chemotherapy is often associated with limited survival benefits and considerable toxicity, resulting in poor quality of life and a median overall survival (OS) of less than three years for endocrine-resistant cases [3,4,5].

Metastatic triple-negative breast cancer (mTNBC), accounting for 10–15% of all breast cancers, exhibits an aggressive clinical course with a high risk of early metastasis and recurrence, particularly to visceral organs and the central nervous system (CNS). Due to the lack of well-defined molecular targets, systemic chemotherapy has remained the cornerstone of treatment for mTNBC, although its outcomes remain suboptimal, with short progression-free survival (PFS) and OS following standard therapies [6,7].

Sacituzumab govitecan (SG) is a novel antibody–drug conjugate that targets the transmembrane glycoprotein Trop-2, which is overexpressed in a wide range of epithelial cancers, including both mTNBC and mHRPBC. SG consists of a humanized anti-Trop-2 monoclonal antibody conjugated to SN-38—the active metabolite of irinotecan—via a hydrolysable linker. This structure allows for targeted delivery of cytotoxic therapy directly into tumor cells and exerts a secondary “bystander effect” on the surrounding tumor microenvironment [8,9,10,11,12].

Following early-phase studies, SG received accelerated approval by the United States Food and Drug Administration (FDA) in 2020 for the treatment of patients with mTNBC who had received at least two prior lines of therapy. Its efficacy was later confirmed in the phase III ASCENT trial, which demonstrated significantly improved survival outcomes and response rates compared to standard chemotherapy [13]. More recently, the TROPiCS-02 trial expanded the use of SG to the mHRPBC setting, establishing its benefit over physician’s choice chemotherapy in heavily pretreated, endocrine-resistant patients [14].

While randomized clinical trials have validated the efficacy and safety of SG, real-world data on its use remain relatively limited, particularly among diverse patient populations and across both major Trop-2-expressing subtypes. Observational studies reflecting clinical practice are essential to understanding how SG performs outside of controlled trial settings, especially with respect to tolerability, treatment patterns, and prognostic factors [15,16,17,18,19].

The present multicenter, retrospective study aims to describe and compare the real-world clinical outcomes, tolerability, and prognostic indicators associated with SG in patients with mTNBC and mHRPBC. By including both molecular subtypes, this analysis seeks to provide a comprehensive overview of clinical outcomes and identify factors associated with treatment response and survival in a real-life clinical setting.

## 2. Materials and Methods

### 2.1. Study Design and Population

This retrospective, multicenter study included 68 patients diagnosed with mTNBC or mHRPBC, treated with SG between 2022 and 2025. The study was conducted across several oncology centers, and only female patients aged 18 and above were included. For inclusion in the mTNBC cohort, patients were required to have experienced progression after at least one line of chemotherapy or chemoimmunotherapy, whereas mHRPBC patients were included if they had previously received at least two lines of chemotherapy that failed prior to CDK 4/6 inhibitors and other hormonotherapy agents. Patients were excluded if they had received SG as part of a clinical trial, had other primary cancers within the past 5 years (excluding non-metastatic skin cancers), did not provide informed consent, or were male. As this was a retrospective, observational study without a control or comparator group, its design was intended not to assess treatment efficacy but rather to describe clinical outcomes and tolerability in a real-world setting.

### 2.2. Intervention and Treatment Protocol

Sacituzumab govitecan was administered at a dose of 10 mg/kg on days 1 and 8 of a 21-day cycle. In cases of dose reduction, the dosage was decreased to 7.5 mg/kg, followed by 5 mg/kg based on patient tolerance. Concomitant granulocyte colony-stimulating factor (G-CSF) was administered as primary prophylaxis in the majority of patients to mitigate the risk of neutropenia. The treatment continued until disease progression, unacceptable toxicity, or the patient’s decision to discontinue.

### 2.3. Outcome Measures

The primary study outcomes included PFS and OS. PFS was defined as the time from the initiation of SG treatment to disease progression or death from any cause. OS was also calculated from the initiation of SG treatment to death from any cause. Safety assessments included the incidence and severity of adverse events (AEs), which were classified according to the National Cancer Institute Common Terminology Criteria for Adverse Events (NCI-CTCAE), version 5.0. Dose reductions and treatment discontinuations due to toxicity were also recorded.

### 2.4. Statistical Analysis

A statistical analysis was performed using IBM SPSS Statistics for Windows, Version 27.0 (IBM Corp., Armonk, NY, USA) Descriptive statistics were used to summarize the baseline demographic and clinical characteristics of the patients, with categorical variables expressed as frequencies and percentages and continuous variables as medians with ranges or means with standard deviations.

The primary outcomes, PFS and OS, were estimated using the Kaplan–Meier method. The OS was calculated from the initiation of SG treatment to death from any cause. The differences in survival between groups were evaluated using the log-rank test (Mantel–Cox test) for comparing the survival distributions across different subgroups, such as molecular subtypes (mTNBC vs. mHRPBC), Eastern Cooperative Oncology Group Performance Status (ECOG PS-0 vs. ECOG PS-1), and metastatic sites (liver, bone, brain, etc.).

A univariate analysis of PFS and OS was conducted to assess the impact of baseline variables, including molecular subtype, ECOG PS, metastatic sites, previous treatment regimens, and Ki-67 expression, on survival outcomes. Factors identified as significant in the univariate analysis were included in the multivariate Cox regression model to determine independent predictors of survival. Hazard ratios (HRs) with 95% confidence intervals (CIs) were calculated for each predictor. Additionally, binary logistic regression was performed to evaluate factors associated with treatment response. Variables with significant associations in the univariate analysis were included in the logistic regression model. All statistical tests were two-tailed, and a *p*-value of less than 0.05 was considered statistically significant.

### 2.5. Ethics Statement

This study was approved by the relevant institutional review boards at each participating center. Ethical approval was granted by the Istanbul Medipol University Ethics Committee (Ethics Committee Decision No: 1209, Date: 28 November 2024). Informed consent was obtained from all participants before the initiation of SG treatment.

## 3. Results

### 3.1. Baseline Characteristics

The baseline clinical characteristics of the 68 patients included in the study are summarized in Table 1. The median age was 48 years (range: 29–78). The majority of patients had an ECOG PS of 0 (70.6%) and received prior taxane (94.1%) and anthracycline (79.4%) therapies. De novo metastatic disease was present in 26.5% of the patients. In terms of molecular classification, 51.5% had mTNBC, and 48.5% had mHRPBC. The HER2 status was predominantly immunohistochemistry (IHC) 0 (70.4%). In the metastatic setting, 42.6% of the patients had received three or fewer prior lines of therapy, while 55.9% had received more than three lines. The most frequent sites of metastasis were lymph nodes (85.3%), bone (57.4%), and lung (57.4%), followed by liver (51.5%) and brain (42.6%). The median number of SG cycles was 7 (range: 3–37), and primary prophylaxis with G-CSF support was used in 88.2% of cases.

### 3.2. Clinical Outcomes

At a median follow-up of 6.8 months (95% CI: 5.4–10.0), the median PFS was 6.1 months (95% CI: 4.83–7.43), and the median mOS was 12.5 months (95% CI: 9.92–15.07) in the entire cohort (Figure 1 and Figure 2, respectively).

There was no significant difference observed between the molecular subgroups, with PFS durations of 6.5 months for mTNBC and 5.76 months for mHRPBC (*p* = 0.78) (Figure 3). Similarly, no significant difference in PFS was found for the de novo metastasis group (*p* = 0.63). However, ECOG PS had a significant impact on PFS, with patients in ECOG PS-0 having significantly longer PFS compared to those in ECOG PS-1 (*p* = 0.004).

Regarding metastatic sites, liver metastasis significantly shortened the PFS (*p* = 0.002). Bone metastasis also had a significant effect on PFS (*p* = 0.004), whereas no significant difference was observed for lung metastasis (*p* = 0.088) or brain metastasis (*p* = 0.253). Similarly, lymph node metastasis did not significantly affect PFS (*p* = 0.086).

Having received immunotherapy before did not make a significant difference (*p* = 0.886). There were no significant differences observed for different chemotherapy histories and treatment types. Taxanes, anthracyclines, carboplatin, and capecitabine did not show any significant impact on PFS. Additionally, factors such as local treatment and the number of chemotherapy lines did not result in significant differences in PFS.

In the analysis of Ki-67 values, no significant difference was observed between patients with Ki-67 ≤ 20 and those with Ki-67 > 20 (*p* = 0.897). Moreover, no relationship was found between Ki-67 values in the metastatic setting and PFS. Finally, dose reduction due to toxicity and G-CSF use did not have a significant effect on PFS (*p* = 0.270 and *p* = 0.097, respectively).

These findings suggest several factors that may be associated with patient prognosis. Specifically, ECOG PS, liver metastasis, and bone metastasis have significant effects on PFS, while treatment types and Ki-67 values do not show a meaningful impact on PFS (Table 2).

The multivariate analysis evaluated independent prognostic factors for PFS. It found that liver metastasis significantly increased the risk of disease progression (*p* = 0.047, HR = 2.046), with faster disease progression observed in these patients. Similarly, ECOG PS was associated with faster disease progression and shorter PFS (*p* = 0.050, HR = 1.968). However, bone metastasis and the molecular subtype did not show a significant impact on PFS (*p* = 0.095 and *p* = 0.348, respectively) (Table 2).

Furthermore, in the univariate analysis, molecular subgroups such as mTNBC and mHRPBC did not show a significant difference in OS (*p* = 0.38), with median OS values of 11.93 months (95% CI: 5.22–18.64 months) for mTNBC and 11.3 months (95% CI: 9.16–25.4 months) for mHRPBC (Figure 4).

In the de novo metastatic cohort, the median OS was 14.73 months (95% CI: 4.23–25.23 months), with no significant difference (*p* = 0.716). ECOG PS also did not significantly impact OS (*p* = 0.178).

Regarding metastatic sites, liver metastasis was strongly associated with worse OS (5.96 months, *p* = 0.001), as were bone and brain metastases, with median OS values of 11.93 months (*p* = 0.008) and 7.13 months (*p* = 0.02), respectively. Lung metastasis did not significantly affect OS (*p* = 0.076). Treatment modalities showed varied impacts; carboplatin significantly improved OS (12.30 months, *p* = 0.04), while other treatments, including immunotherapy and anthracyclines, showed no significant difference (*p* > 0.05).

The number of chemotherapy lines, treatment discontinuation due to toxicity, and G-CSF use did not show significant effects on OS. Ki-67 expression also did not show a significant difference in OS, either at initial diagnosis or in the metastatic setting (Table 3).

When a multivariate analysis was carried out for OS, liver metastasis was identified as an independent predictor of poorer OS (HR = 3.150, 95% CI: 1.184–8.383, *p* = 0.022). Bone metastasis showed a trend toward statistical significance (HR = 2.283, 95% CI: 0.927–5.624, *p* = 0.073). The molecular subtype was also significantly associated with OS, as mTNBC patients had better outcomes compared to mHRPBC patients (HR = 0.467, 95% CI: 0.221–0.987, *p* = 0.046).

Brain metastasis did not show a significant impact on OS (HR = 1.398, 95% CI: 0.609–3.205, *p* = 0.429). Overall, liver metastasis and the molecular subtype were independent predictors of poorer overall survival (Table 3).

Among the 68 evaluable patients, the objective response rate (ORR) was 52.9%, with 7 patients (10.3%) achieving a complete response (CR) and 29 patients (42.6%) achieving a partial response (PR). Additionally, stable disease (SD) was observed in 14.7% of cases, resulting in a disease control rate (DCR) of 67.6%.

A binary logistic regression analysis was performed to identify the factors associated with treatment response. As shown in Figure 5, liver metastasis (Odds Ratio (OR): 6.49, *p* = 0.038) and lung metastasis (OR: 7.59, *p* = 0.013) were significantly associated with a higher likelihood of treatment response, with patients having approximately 6.5 and 7.5 times higher odds of response, respectively. Bone metastasis showed a borderline association (OR: 4.35, *p* = 0.050), suggesting a 4-fold increased likelihood of response. However, lymph node metastasis was associated with a markedly lower likelihood of response (OR: 0.065, *p* = 0.017), indicating a significantly reduced chance of treatment efficacy. Other factors, including de novo disease, molecular subtypes, ECOG PS, brain metastasis, and Ki-67 index values, were not significantly associated with treatment response (all *p*-values > 0.05) (Figure 5).

### 3.3. Safety and Adverse Events

The AE data are visualized in Figure 6. The most common toxicity was alopecia, reported in 90% of patients. Anemia occurred in 41.7% of patients, with 15% experiencing grade ≥ 3 severity. Other frequent hematologic toxicities included neutropenia and thrombocytopenia, with a substantial proportion at grade ≥ 3 severity, highlighting the importance of routine hematologic monitoring and supportive care. Notably, primary prophylaxis with G-CSF was used in 88.2% of patients to manage neutropenia, emphasizing the role of G-CSF in mitigating hematologic toxicity associated with SG treatment.

Among the non-hematologic toxicities, nausea and diarrhea were the most common. For patients experiencing grade 2 or higher nausea and vomiting, dexamethasone, 5-hydroxytryptamine 3 receptor agonists (5-HT3), and neurokinin-1 receptor antagonist (NK-1) inhibitors were routinely administered as part of the antiemetic regimen. Diarrhea was managed with anti-diarrheal agents when patients experienced grade 2 or higher severity, reflecting a proactive approach to managing gastrointestinal toxicity. Fatigue was reported in 21.7% of patients, with a small proportion experiencing higher-grade fatigue. Elevated liver enzymes and febrile neutropenia were infrequent and did not require a treatment interruption.

In terms of treatment modifications due to toxicity, a dose reduction occurred in 29.4% of patients, and treatment discontinuation due to toxicity was observed in 2.9% of the cohort, which was attributed to treatment-refractory grade 4 thrombocytopenia. These findings underscore the importance of early recognition and management of adverse events, as they can significantly impact treatment continuity and efficacy.

## 4. Discussion

### 4.1. Real-World Outcomes Compared to Clinical Trials

This retrospective, multicenter, real-world study descriptively reports the clinical course and tolerability outcomes of SG treatment in patients with mTNBC and mHRPBC in routine clinical practice in Turkey. While the median PFS and OS in this cohort appear similar to those reported in clinical trials, these results should not be interpreted as confirmation of treatment efficacy. These findings offer descriptive, observational insights into how SG performs across two distinct biological subtypes in a real-world setting [20,21].

In our cohort, the observed median PFS and OS were 6.1 months and 12.5 months, respectively. These outcomes appear similar to those reported in pivotal clinical trials. For example, the ASCENT study reported a median PFS of 5.6 months and OS of 12.1 months in pretreated mTNBC patients [13], while TROPiCS-02 demonstrated a median PFS of 5.5 months and OS of 14.4 months in mHRPBC patients [14]. The EVER-132-002 trial focusing on Asian patients also confirmed these outcomes, with a median OS of 21.0 months in the SG arm [22]. While the median PFS and OS in this cohort appear similar to those reported in clinical trials, these outcomes should be interpreted descriptively due to the observational design and absence of a comparator arm.

A particularly novel contribution of our study is the inclusion of mHRPBC patients, for whom real-world data are remarkably sparse. To our knowledge, this is the first real-life cohort to systematically evaluate SG in both mTNBC and mHRPBC populations. Despite the growing body of evidence supporting SG use in mHRPBC disease, from trials such as TROPiCS-02 and EVER-132-002, there remains a significant gap in the observational literature. Previously published real-world studies have almost exclusively focused on mTNBC, leaving a critical evidence gap for mHRPBC patients [16,23].

In our study, 33 patients (48.5% of the cohort) had mHRPBC and had previously received CDK 4/6 inhibitors plus endocrine therapy and at least two lines of chemotherapy. In this subgroup, the median PFS was 6.1 months and the median OS was 12.5 months, which are consistent with the outcomes reported in the TROPiCS-02 trial (median PFS 5.5 months, OS 14.4 months) [14]. These real-world findings support the therapeutic role of SG in chemotherapy-pretreated mHRPBC patients and demonstrate comparable effectiveness to that in clinical trial populations.

Our findings are further supported by the EVER-132-002 study, which reported a favorable safety and efficacy profile of SG in Asian mHRPBC patients [22]. However, to our knowledge, no real-world studies have provided detailed clinical outcome data specifically for the mHRPBC subgroup. Therefore, the current analysis contributes to the existing literature by providing limited real-world data on SG use in mHRPBC, offering insights that may help inform treatment strategies for endocrine-resistant, chemotherapy-pretreated patients.

Our analysis identified two major clinical parameters—ECOG PS ≥ 1 and the presence of liver metastases—as independent predictors of shorter PFS and OS. These findings are consistent with previous real-world studies and subgroup analyses of clinical trials that have reported associations between poor ECOG PS, visceral involvement, and inferior clinical outcomes [24,25,26]. In our multivariate model, patients with ECOG PS ≥ 1 experienced significantly shorter OS, highlighting the prognostic importance of baseline functional status.

Liver metastases, in particular, were a consistent marker of poor prognosis in our study, mirroring findings from both the ASCENT trial and a meta-analysis that highlighted diminished SG benefit in patients with hepatic involvement [25]. This is further supported by Italian and Polish real-world cohorts, wherein patients with liver metastases also showed worse clinical outcomes [16,23].

Intriguingly, despite their association with worse survival, liver and lung metastases in our cohort were independently associated with a higher ORR. This seemingly paradoxical finding may reflect an initial tumor burden reduction before progression or enhanced drug delivery in highly perfused organs. It is also possible that such patients benefit from early disease control but eventually succumb to aggressive tumor biology or resistance mechanisms. Further molecular and pharmacokinetic studies are needed to explore this observation.

Another notable subgroup within our population consisted of patients with CNS metastases. Although SG is not formally approved for the treatment of active brain metastases, observational data, including ours, indicate that stable or previously treated CNS disease does not necessarily preclude a clinical benefit with SG [17,18]. In our cohort, the presence of brain metastases was not significantly associated with worse PFS or OS. However, due to the study’s observational design and the small number of patients with CNS involvement, these findings should be interpreted cautiously. Further controlled studies are warranted to better define the role of SG in patients with CNS metastases.

In addition to these subgroups, a relevant clinical consideration is the impact of prior immunotherapy exposure. A subset of our patients—primarily within the mTNBC cohort—had received immune checkpoint inhibitors (ICIs) prior to initiating SG. In our cohort, prior ICI exposure did not appear to be associated with worse clinical outcomes following SG treatment. This observation is consistent with findings from the ASCENT trial, where prior ICI use did not negatively impact clinical benefits [13]. Furthermore, emerging real-world data suggest that SG remains a treatment option for patients previously exposed to immunotherapy [23,27,28]. Additional prospective studies are needed to better define the impact of prior immunotherapy on SG outcomes.

Together, these subgroup analyses contribute important granularity to patient selection in the real-world setting. They also highlight the need for further biomarker-guided stratification to identify individuals who derive the most benefit from SG, especially in heavily pretreated and clinically diverse populations.

The ORR was 52.9%, with a CR rate of 10.3%. This is higher than those in previous real-world reports: for instance, an Italian cohort showed an ORR of 33.3% [23], and a US cohort reported 27.8% [19]. These differences may reflect variations in baseline patient characteristics, disease burdens, treatment adherence, or supportive care practices, including the high G-CSF utilization in our population.

### 4.2. Safety and Tolerability Profile

In our study, SG was generally well tolerated, with AEs consistent with those reported in clinical trials and prior real-world studies. The most frequently observed AEs included alopecia (64.7%), anemia (52.9%), neutropenia (50%), and diarrhea (38.2%). Grade ≥ 3 hematologic toxicities, especially neutropenia, were observed in a substantial proportion of patients but were successfully managed with G-CSF, which was administered to 88.2% of patients. Only 2.9% of patients discontinued treatment due to AEs, and no new safety signals emerged.

These findings are consistent with the established safety profile of SG as seen in the ASCENT [13] and TROPiCS-02 [14] trials. Furthermore, they align with findings from other real-world analyses. For instance, an Italian multicenter study reported anemia (66.6%), neutropenia (59.6%), and diarrhea (38.6%) as the most common toxicities, with treatment discontinuation due to AEs in 5.3% of patients [23]. Similarly, German [17] and Polish [16] cohorts reported comparable safety profiles, suggesting that SG-associated toxicity is generally predictable and manageable with supportive care.

A recent meta-analysis further supported these observations, concluding that although SG is associated with a higher incidence of grade 3–4 neutropenia and anemia compared to standard chemotherapy, it does not significantly increase the risk of treatment discontinuation [25].

Given the high rate of G-CSF usage in our cohort, the role of proactive supportive care should not be underestimated. Early recognition and management of hematologic toxicity are important for maintaining the dose intensity and minimizing treatment interruptions. Our findings highlight the potential importance of preemptive strategies to mitigate toxicity-related disruptions in therapy.

### 4.3. Limitations

This study has several limitations, some of which are related to contextual and regulatory factors specific to the study setting. First, its retrospective design may have introduced bias in data collection and interpretation. Second, the relatively small sample size may have limited the statistical power, particularly for subgroup analyses. Third, the study did not include analyses of biomarkers such as Trop-2 expression, which may provide additional predictive information. Fourth, treatment selection and supportive care measures were not standardized across the centers. Importantly, the absence of a control or comparator group precludes definitive conclusions regarding the effectiveness of SG. The observed clinical outcomes must therefore be interpreted descriptively rather than as evidence of treatment efficacy. Additionally, the relatively low number of patients included in the study was influenced by access-related limitations. At the time this study was initiated, SG was not reimbursed by national health authorities for patients with mHRPBC in Turkey. Even among mTNBC patients, reimbursement was restricted to those who had received at least two prior chemotherapy regimens in the metastatic setting. These reimbursement constraints likely delayed treatment initiation and limited broader access to SG, thereby reducing patient enrollment and potentially affecting the generalizability of the findings.

### 4.4. Clinical Implications

In conclusion, this study provides real-world observational data on the use of SG in patients with mTNBC and mHRPBC treated in routine clinical practice. Importantly, our analysis highlights the paucity of real-world data in the mHRPBC subgroup, contributing valuable information to the existing literature. Although our real-world findings mirror clinical trial outcomes in terms of PFS and OS, the absence of a control arm necessitates a cautious interpretation regarding treatment impacts. These results suggest that SG can be feasibly administered across both subtypes, particularly in heavily pretreated, endocrine-resistant patients. Further prospective, controlled, and biomarker-driven studies are warranted to validate these observations and optimize patient selection, sequencing strategies, and supportive care approaches.

## 5. Conclusions

This multicenter, retrospective real-world study provides descriptive insights into the clinical outcomes and tolerability of SG in patients with mTNBC or mHRPBC. Our findings indicate that PFS and OS outcomes were broadly similar across the two molecular subtypes, suggesting comparable real-world clinical trajectories under SG treatment.

Importantly, this study addresses a critical gap in the literature by offering one of the first real-world datasets on SG use in the mHRPBC population, a subgroup that has been underrepresented outside of clinical trials. Despite previous endocrine therapy and multiple lines of chemotherapy, patients with mHRPBC demonstrated outcomes similar to those observed in mTNBC, underscoring the potential role of SG in heavily pretreated, endocrine-resistant disease.

Prognostic factors such as ECOG PS and liver metastasis were identified as significant determinants of survival, reinforcing the importance of baseline clinical characteristics in guiding treatment decisions. Additionally, the observed tolerability profile was consistent with clinical trial data, with manageable toxicity rates and a low incidence of treatment discontinuation.

In summary, SG appears to be a feasible therapeutic option across both mTNBC and mHRPBC subtypes in routine clinical practice. Future prospective, controlled studies with biomarker-based stratification are warranted to optimize patient selection strategies, particularly within the evolving treatment landscape of mHRPBC.

## Figures and Tables

**Figure 1 cancers-17-01592-f001:**
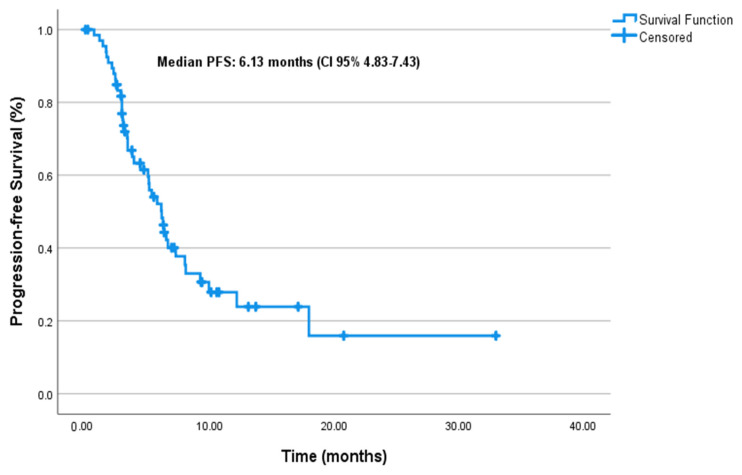
Progression-free survival.

**Figure 2 cancers-17-01592-f002:**
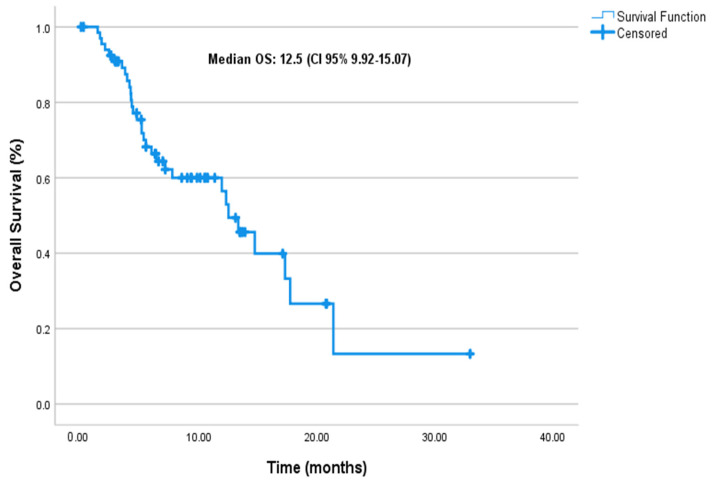
Overall survival.

**Figure 3 cancers-17-01592-f003:**
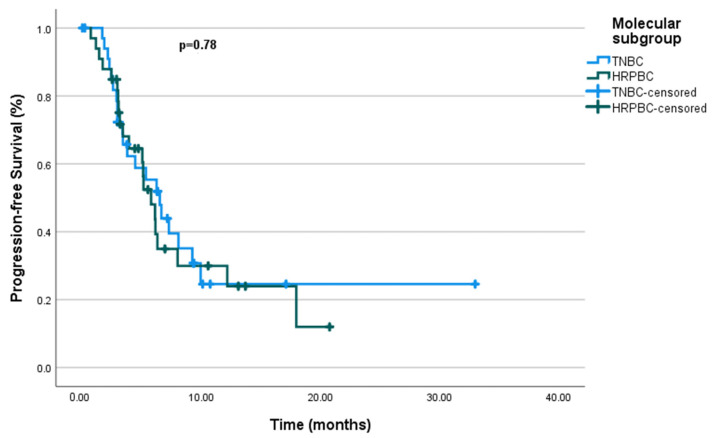
Progression-free survival by molecular subgroup.

**Figure 4 cancers-17-01592-f004:**
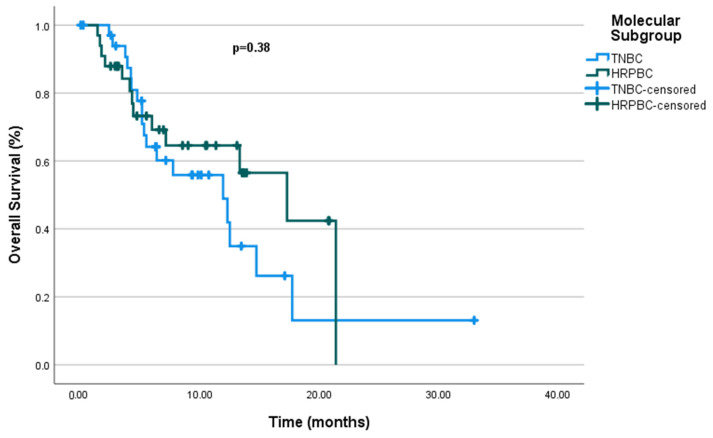
Overall survival by molecular subgroup.

**Figure 5 cancers-17-01592-f005:**
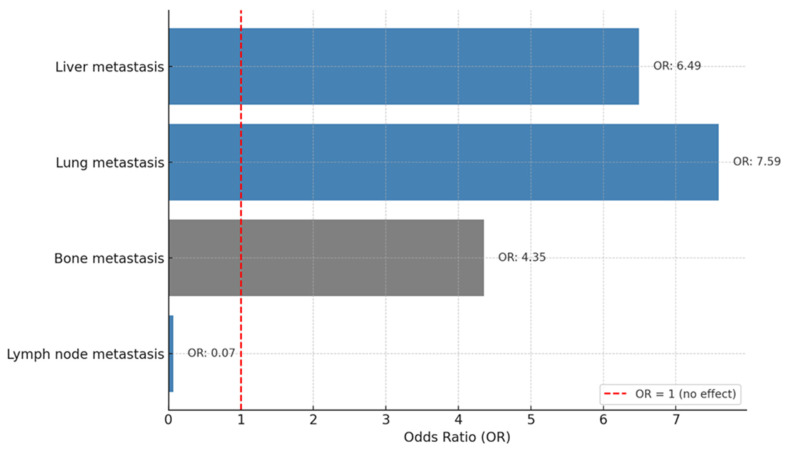
Factors associated with treatment response.

**Figure 6 cancers-17-01592-f006:**
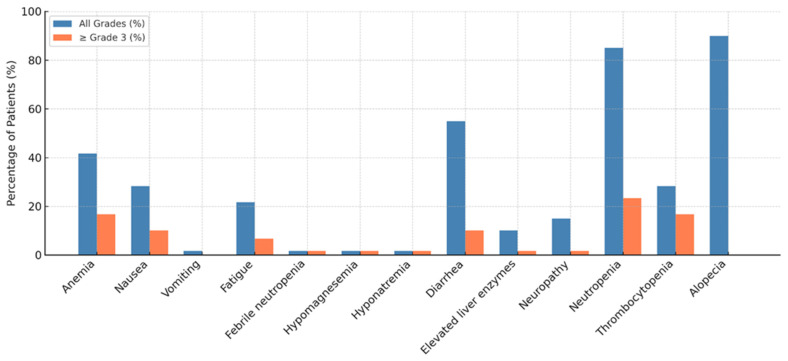
Adverse events.

**Table 1 cancers-17-01592-t001:** Baseline characteristics.

Variable	Value
**Age (median, range)**	48 (29–78)
**De novo metastasis**	18 (26.5%)
**Molecular classification**	
mTNBC	35 (51.5%)
mHRPBC	33 (48.5%)
**Her2 status**	
Her2 0	52 (76.4%)
Her2 + 1	10 (14.7%)
Her2 + 2 (FISH negative)	6 (8.8%)
**ECOG PS**	
0	48 (70.6%)
1	20 (29.4%)
**Metastatic sites**	
Liver	35 (51.5%)
Lung	39 (57.4%)
Brain	29 (42.6%)
Bone	39 (57.4%)
Lymph node	58 (85.3%)
**Prior immunotherapy**	22 (32.4%)
**Dose reduction due to toxicity**	20 (29.4%)
**Treatment discontinuation due to toxicity**	2 (2.9%)
**Prior chemotherapy agents**	
Taxane	64 (94.1%)
Anthracycline	54 (79.4%)
Carboplatin	48 (70.6%)
Capecitabine	53 (77.9%)
Local treatment	60 (88.2%)
**Prior lines of therapy in metastatic set**	
≤3 lines	29 (42.6%)
>3 lines	38 (55.9%)
**Number of SG cycles (median, range)**	7 (3–37)
**G-CSF use with SG**	60 (88.2%)

mTNBC: metastatic triple-negative breast cancer, mHRPBC: metastatic hormone receptor-positive/human epidermal growth factor receptor 2-negative breast cancer, ECOG PS: Eastern Cooperative Oncology Group Performance Status, FISH: fluorescence in situ hybridization, SG: sacituzumab gavitecan, G-CSF: granulocyte colony-stimulating factor.

**Table 2 cancers-17-01592-t002:** Univariate and multivariate analyses for progression-free survival.

Variable	mPFS (Months)	95% CI	*p*-Value	HR (95% CI)	Multivariate *p*-Value
**Molecular subgroup**			0.78		0.348
mHRPBC	5.76	(4.28–7.24)		Ref.	
mTNBC	6.5	(4.45–8.54)		0.73 (0.384–1.401)	
**De novo metastases**			0.63		
Absent	6.13	(4.71–7.88)			
Present	5.13	(1.97–8.28)			
**ECOG PS**			**0.004**		**0.050**
ECOG PS-0	7.26	(5.32–9.21)		Ref.	
ECOG PS-1	3.76	(2.26–5.27)		1.96 (0.999–3.875)	
**Liver metastases**			**0.002**		**0.047**
Absent	NR	(4.83–7.43)		Ref.	
Present	4.43	(2.74–6.12)		2.04 (1.008–4.151)	
**Lung metastases**			0.088		
Absent	8.0	(6.19–9.80)			
Present	3.9	(2.30–7.76)			
**Brain metastases**			0.253		
Absent	6.50	(5.44–7.55)			
Present	5.03	(2.74–6.12)			
**Bone metastases**			**0.004**		0.095
Absent	NR	NA		Ref.	
Present	5.03	(3.18–6.88)		1.87 (0.89–3.91)	
**Lymph node metastases**			0.086		
Absent	5.33	(4.31–6.35)			
Present	6.50	(4.07–8.92)			
**Prior ICIs**			0.886		
Absent	6.10	(4.89–7.30)			
Present	6.30	(2.31–10.28)			
**Prior chemotherapy**			0.352		
Taxane	6.13	(4.86–7.40)			
Antracycline	6.13	(4.79–7.47)			
Carboplatin	6.23	(4.62–7.84)			
Capecitabine	6.13	(4.94–7.32)			
**Local treatment**			0.929		
Absent	3.40	(0.10–11.71)			
Present	6.13	(4.90–7.36)			
**No. of chemotherapy lines**			0.796		
≤3 lines chemotherapy	5.33	(2.89–7.76)			
>3 lines chemotherapy	6.23	(5.32–7.14)			
**Dose reduction due to toxicity**			0.270		
Absent	6.23	(5.33–7.13)			
Present	3.13	(0.13–6.13)			
**G-CSF use with SG**			0.097		
Absent	NR	NA			
Present	NR	NA			
**At diagnosis Ki-67**			0.897		
≤20	6.13	(4.55–7.71)			
>20	5.76	(4.27–7.25)	0.897		
**Metastatic setting Ki-67**			1		
≤20	6.13	(3.46–8.80)			
>20	6.23	(4.67–7.79)	1		

mTNBC: metastatic triple-negative breast cancer, mHRPBC: metastatic hormone receptor-positive/human epidermal growth factor receptor 2-negative breast cancer, ECOG PS: Eastern Cooperative Oncology Group Performance Status, G-CSF: granulocyte colony-stimulating factor, ICIs: immune checkpoint inhibitors, HR: hazard ratio, CI: confidence interval, NR: not reached, NA: not applicable, mPFS: median progression-free survival, Ref: reference category.

**Table 3 cancers-17-01592-t003:** Univariate and multivariate analyses for overall survival.

Variable	mOS (Months)	95% CI	*p*-Value	HR (95% CI)	Multivariate *p*-Value
**Molecular subgroup**			0.380		**0.046**
mHRPBC	11.30	(9.16–25.4)		Ref.	
mTNBC	11.93	(5.22–18.64)		0.46 (0.22–0.98)	
**De novo metastases**			0.716		
Absent	12.50	(10.78–14.21)			
Present	14.73	(4.23–25.23)			
**ECOG-PS**			0.178		
ECOG-0	14.73	(10.52–18.94)			
ECOG-1	13.33	(2.56–24.10)			
**Liver metastases**			**0.001**		**0.022**
Absent	17.73	NA		Ref.	
Present	5.96	(2.79–9.14)		3.15 (1.184–8.383)	
**Lung metastases**			0.076		
Absent	17.30	(11.29–23.31)			
Present	7.13	(3.65–14.78)			
**Brain metastases**			**0.025**		0.429
Absent	17.30	(11.12–23.47)		Ref.	
Present	7.13	(1.07–13.18)		1.39 (0.609–3.205)	
**Bone metastases**			**0.008**		0.073
Absent	NR	NA		Ref.	
Present	11.93	(4.98–18.88)		2.28 (0.927–5.624)	
**Lymph node metastases**			0.884		
Absent	21.40	NA			
Present	12.50	(10.72–14.27)			
**Prior ICIs**			0.963		
Absent	14.73	(4.89–24.57)			
Present	12.50	(10.31–14.68)			
**Prior chemotherapy**			0.293		
Taxane	-	-			
Antracycline	13.33	(4.56–22.10)			
Carboplatin	12.30	(4.71–19.88)			
Capecitabine	12.50	(9.95–15.04)			
**Local treatment**			0.673		
Absent	6.36	(0.10–16.03)			
Present	12.50	(10.75–14.24)			
**No. of chemotherapy lines**			0.745		
≤3 lines chemotherapy	14.73	(4.99–24.47)			
>3 lines chemotherapy	12.30	(5.81–18.79)			
**Dose reduction due to toxicity**			1.00		
Absent	12.50	(10.70–14.29)			
Present	NR	NA			
**G-CSF use with SG**			0.724		
Absent	12.30	(0.10–24.75)			
Present	13.33	(6.96–19.69)			
**At diagnosis Ki-67**			0.460		
≤20%	14.73	(4.85–22.63)			
>20%	12.30	(6.02–18.57)			
**Metastatic setting Ki-67**			0.184		
≤20%	14.73	(2.81–30.21)			
>20%	12.50	(10.82–14.17)			

mTNBC: Metastatic triple-negative breast cancer, mHRPBC: Metastatic hormone reseptore positive/Human epidermal growth factor receptor 2 negative breast cancer, ECOG PS: Eastern cooperative oncology group performance status, G-CSF: Granulocyte colony-stimulating factor, ICIs: Immune check point inhibitors, HR: Hazard ratio, CI: Confidence interval, NR: Not reached, NA: Not applicable, mOS: Median overall survival, Ref: Reference category.

## Data Availability

The data from this study are available from the corresponding author upon reasonable request.

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
