# Peer review of "The Clinical Outcomes and Safety of Sacituzumab Govitecan in Heavily Pretreated Metastatic Triple-Negative and HR+/HER2− Breast Cancer: A Multicenter Observational Study from Turkey"

_cancers, 2025, doi:10.3390/cancers17091592_

Round 1
Reviewer 1 Report
Comments and Suggestions for Authors
Authors present a multicenter, retrospective study aimed to evaluate the real-life effectiveness, safety, and prognostic factors associated with Sacituzumab govitecan treatment in patients with metastatic triple-negative breast cancer (mTNBC) and hormone receptor-positive/HER2-negative (mHRPBC) breast cancer. Study includes 68 patients. The median PFS was 6.1 months, and the median OS was 12.5 months, with no significant differences between subtypes. Authors concluded that SG demonstrated comparable effectiveness and a managea-ble safety profile in real-world patients with both mTNBC and mHRPBC.
Authors conducted different analyses and presented tables and figures with interesting results.
However, I see only one but important major issue.
Authors compared the same therapy between two cancer types. They can say that the effectiveness and side effects of SG was similar in both cancer types. But they cannot say “Our findings confirm and extend the clinical trial data by demonstrating that SG retains its efficacy and manageable safety profile in real-world populations”. Why? To show the effectiveness of the therapy, they need to compare this therapy with no therapy (in clinical trial this would be placebo, in real world setting this would be another therapy like chemo or no therapy). Authors show PFS of 6.1 months? Is it much or is it less? It looks too short time; but would women here live several months less when they would not get SG? We do not see it in this study. So the conclusion should be changed, no messages about effectiveness are possible. The only conclusion here, and probably the study goal can be redefined too, is to show and to compare the outcomes between two cancer types.
Reviewer 2 Report
Comments and Suggestions for Authors
Tables
The HR:
Hazard ratio is not applied to all vaiarble-- applied on groups-- so better to make row demarcation .
The title:
Advised to make more shorter-- example the Real World word can be removed
"The Efficacy and Safety of Sacituzumab Govitecan in Heavily Pretreated Metastatic Triple-Negative and Hormone Receptor Positive/HER2-Negative Breast Cancer: A Multicenter Study from Turkey"
Reviewer 3 Report
Comments and Suggestions for Authors
This is a well-conducted and well-written study that provides valuable real-world evidence on the efficacy and safety of sacituzumab govitecan in patients with metastatic breast cancer, particularly those with limited treatment options. Including both mTNBC and mHRPBC subtypes enhances the relevance and applicability of the findings. The statistical analyses are robust, and the results are presented comprehensively and coherently. I found the manuscript scientifically sound and of high quality. I have no further comments or suggestions for improvement.
Round 2
Reviewer 1 Report
Comments and Suggestions for Authors
-